



# Parameter optimisation for a better representation of drought by LSMs: inverse modelling vs. sequential data assimilation

Hélène Dewaele[1], Simon Munier[1], Clément Albergel[1], Carole Planque[1], Nabil Laanaia[1], Dominique Carrer[1], Jean-Christophe Calvet[1]

[1]CNRM – UMR3589 (Météo-France, CNRS), Toulouse, 31057, France

*Correspondence to*: Jean-Christophe Calvet (jean-christophe.calvet@meteo.fr)

**Abstract.** Soil Maximum Available Water Content (MaxAWC) is a key parameter in Land Surface Models (LSMs). However, being difficult to measure, this parameter is usually unavailable. This study assesses the feasibility of using a fifteen-year (1999-2013) time-series of satellite-derived low resolution observations of Leaf Area Index (LAI) to retrieve

MaxAWC for rainfed croplands over France. LAI inter-annual variability is simulated using the $CO_2$-responsive version of the Interactions between Soil, Biosphere and Atmosphere (ISBA) LSM for various values of MaxAWC. Optimal value is then selected by using (1) a simple inverse modelling technique, comparing simulated and observed LAI, (2) a more complex method consisting in integrating observed LAI in ISBA through a Land Data Assimilation System (LDAS) and minimizing LAI analysis increments. The evaluation of the MaxAWC retrievals from both methods is done using simulated

annual maximum above-ground biomass ($B_{ag}$) and straw cereal grain yield (GY) values from the Agreste French agricultural statistics portal, for 45 administrative units presenting a high proportion of straw cereals. Significant correlations (p-value < 0.01) between $B_{ag}$ and GY are found for up to 36% and 53% of the administrative units for the inverse modelling and LDAS tuning methods, respectively. It is found that the LDAS tuning experiment gives more realistic values of MaxAWC and maximum $B_{ag}$ than the inverse modelling experiment. Using low resolution LAI observations leads to an underestimation of

MaxAWC and maximum $B_{ag}$ in both experiments. Median annual maximum values of disaggregated LAI observations are found to correlate very well with MaxAWC.

## 1 Introduction

In the context of climate change, there is a need for better supervision of the impacts of droughts on crops and water resources at continental and global scales (Quiroga et al., 2010; Van der Velde et al., 2011; Crow et al., 2012; Bastos et al.,

2014). Large scale modelling of continental surfaces into atmospheric and hydrological models has evolved in recent decades towards Land Surface Models (LSMs) able to simulate the coupling of the water, energy and carbon cycles (Calvet et al., 1998; Krinner et al., 2005; Gibelin et al., 2006). In particular, LSMs are now able to simulate photosynthesis and plant growth. A major source of uncertainty in both LSMs and crop models is the maximum available water content of the soil (MaxAWC). This quantity represents the amount of water stored in the soil at field capacity available for plant transpiration



along the vegetation growing cycle (Portoghese et al., 2008; Piedallu et al., 2011). MaxAWC is constrained by soil parameters and by the plant rooting depth. In regions affected by droughts, MaxAWC is a key driver of the plant response to the climate variability.

Extreme weather conditions markedly affect agricultural production. The large scale inter-annual variability of rainfed crop yields is driven to a large extent by the climate variability. Assigning agricultural statistics to climate data shows the impact of atmospheric conditions on vegetation production. For example, lower temperature in northern Europe tends to shorten the period of crop growth. Conversely, persistent high temperatures as well as droughts in southern Europe are linked to negative anomalies of crop yields (Olesen et al., 2011). Li et al. (2010) showed that air temperature tends to influence crop yields at large scales whereas rainfall drive crop yields at smaller scales. Capa-Morocho et al. (2014) also showed the influence of air temperature on crop yields. They established a link between temperature anomalies related to the El Niño phenomenon and potential crop yield anomalies, obtained from reanalysis data and crop model, respectively.

Soil characteristics have an impact on the vegetation response to climate (Folberth et al., 2016). In the model benchmarking study of Eitzinger et al. (2004), simulated evapotranspiration, soil moisture and biomass were compared with observations. They used three crop models differing in the representation of the Available soil Water Content (AWC): WOFOST (WOrld Food Studies model) (Van Diepen et al., 1989), CERES (Crop Environment REsource Synthesis model) (Ritchie and Otter, 1985) and SWAP (Statewide Agricultural Production model) (van Dam et al., 1997). They showed that a better description of rooting depth and evapotranspiration, taking into account soil type and crop type, could significantly improve these models. Other studies have also highlighted the important role of the parameterization of the soil characteristics (soil texture, rooting depth) that determine the water retention capacity of the soil (Tanaka et al., 2004 ; Portoghese et al., 2008 ; Piedallu et al., 2011) but also evapotranspiration (Soylu et al., 2011 ; Wang et al., 2012). While soil properties such as soil texture determine the volumetric soil water holding capacity, information on rooting depth is needed to determine MaxAWC, in units of kg m$^{-2}$. A better representation of MaxAWC could improve the simulated inter-annual variability of both water fluxes and vegetation biomass by LSMs.

The lack of in situ observations of MaxAWC to calibrate and assess LSMs is a significant issue. A list of atmospheric, oceanic and terrestrial Essential Climate Variables (ECVs) which can be monitored at a global scale from remote sensing observations, was proposed by the Global Climate Observing System (GCOS). Leaf area Index (LAI), Fraction of absorbed photosynthetically active radiation (FAPAR) and soil moisture are key ECVs for land surface modelling. The use of these satellite-derived products to verify LSM simulations or to optimize key LSM parameters has been assessed by several authors (e.g. Becker-Reshef et al., 2010 ; Crow et al., 2012 ; Ferrant et al., 2014 ; Ford et al., 2014 ; Ghilain et al., 2012 ; Ichii et al;, 2009 ; Kowalik et al. 2009 ; Szczypta et al., 2012 ; Szczypta et al., 2014). Besides, data assimilation is a field of active research. Data assimilation techniques allow the integration of different observation types (e.g. in situ or satellite-derived) into LSMs in order to optimally combine them with model outputs: the correction applied to the model state is called the increment and the corrected model state is the analysis. In particular, the assimilation of LAI observations can significantly improve the representation of vegetation growth (e.g. Albergel et al., 2010 ; Barbu et al., 2011, 2014).





The Interactions between Soil, Biosphere and Atmosphere (ISBA) LSM includes a modelling option able to simulate photosynthesis and plant growth (Calvet et al., 1998 ; Gibelin et al., 2006). ISBA produces consistent surface energy, water and carbon fluxes, together with key vegetation variables such as LAI and the living above-ground biomass ($B_{ag}$). Previous studies showed that this model can represent well the inter-annual variability of $B_{ag}$ over grassland and straw cereal sites in

France provided MaxAWC values are tuned (Calvet et al., 2012 ; Canal et al., 2014). In these studies, MaxAWC for straw cereals was retrieved by maximizing the correlation coefficient between simulated annual maximum $B_{ag}$ ($B_{agX}$) and grain yield (GY) observations. The MaxAWC values were obtained for 45 French administrative units ("départements") presenting a large proportion of rainfed straw cereals. For grasslands, dry matter yield observations were used. Significant correlations were found between the simulated $B_{agX}$ value of grassland and dry matter yield of grasslands for up to 90 % of the

administrative units. On the other hand, no more than 27 % of the 45 straw cereals départements (i.e. no more than 12 départements) presented significant correlations. A possible cause of the difficulty to simulate the interannual variability of straw cereals' GY was that the standard deviation of GY represented less than 10 % of the mean GY. This was a relatively weak signal. For grasslands a much larger value, of about 30 % of the mean dry matter yield, was observed (Canal et al., 2014).

The main purpose of this study is to retrieve MaxAWC for straw cereals using reverse modelling techniques based on satellite-derived LAI observations disaggregated over separate vegetation types. Simulated and observed LAI are compared for a 15-year period (1999-2013) over the same 45 agricultural spots used in the previous studies of Calvet et al. (2012) and Canal et al. (2014). We use LAI observations instead of GY to retrieve MaxAWC. The GY observations are used to verify the interannual variability of the simulated $B_{agX}$. This can be considered as an indirect validation of the retrieved MaxAWC.

In a first experiment, we use a simple inverse modelling technique to retrieve MaxAWC together with the mass-based leaf nitrogen content, minimising a cost function based on observed and simulated LAI values. In another experiment, we use a Land Data Assimilation System (LDAS) able to sequentially assimilate LAI observations. In this case, MaxAWC solely is retrieved by minimizing the LAI analysis increments.

The main goals of this study are to (1) assess the usefulness of integrating satellite-derived LAI observations into a LSM, (2)

compare inverse modelling and LDAS techniques, (3) determine MaxAWC values. In the following, inverse modelling and LDAS tuning are referred as IM and LT, respectively.

The observation data sets are described in Section 2, together with the version of ISBA used in this study and the LDAS. Results obtained from both methods are presented in Section 3, analysed and discussed in Section 4. Conclusions and prospects are summed up in Section 5.





## 2 Data

The forcing and validation observations used in this study over the 1999-2013 period are described below. The location of the considered straw cereal spots is presented in Fig. 1.

### 2.1. Satellite LAI product

We use the GEOV1 global LAI product (Baret et al., 2013) provided in near real time (every 10 days) at a spatial resolution of 1 km × 1 km by the European Copernicus Global Land Service (http://land.copernicus.eu/global/). The GEOV1 LAI product is derived from SPOT-VGT satellite observations starting in 1999. The complete 1999-2013 LAI time series comes from SPOT-VGT and is fully homogeneous. Camacho et al. (2013) compared the GEOV1 LAI with in situ LAI observations and with different remote sensing products such as MODIS and CYCLOPES. They highlighted that: "The best accuracy and

precision are observed for the GEOV1 LAI product. GEOV1 provides also very good agreement across the whole range of LAI values, with however only a slight underestimation for the highest values". They give the following scores for GEOV1 LAI with respect to ground observations over 30 crop, grass and forest sites in Europe, Africa and North America: $R^2 = 0.81$, RMSE = 0.74 $m^2m^{-2}$.

The GEOV1 product is a low resolution product (1 km x 1 km). At this spatial scale, it is not possible to isolate pure straw

cereal pixels and it is preferable to disaggregate the LAI (i.e. compute the LAI of each vegetation type) before integrating it into a straw cereal model. We disaggregated the GEOV1 LAI data following the method developed by Carrer et al. (2014), based on a Kalman filtering technique. This method permits separating the individual LAI of different vegetation types that co-exist in a grid pixel and then provides dynamic estimates of LAI for each type of vegetation within the pixel (Munier et al., 2017). The Kalman filter optimally combines satellite LAI data and prior information from the ECOCLIMAP land cover

database (Farroux et al., 2007, Masson et al., 2003). ECOCLIMAP prescribes physiographic parameters (fractional vegetation cover, soil depth, etc.) for several vegetation types including grasslands, forests, and C3 crops like straw cereals. Mean annual LAI cycles per vegetation type from ECOCLIMAP are used as a first guess to partition the GEOV1 LAI every time a new satellite observation is available.

### 2.2. Atmospheric forcing

The global WFDEI dataset (Weedon et al., 2014) is used in this study to drive the ISBA simulations. It provides 3-hourly surface atmospheric variables on a 0.5° × 0.5° grid: air temperature, air humidity, wind speed, atmospheric pressure, solid and liquid precipitation, incoming shortwave and longwave radiation. WFDEI is based on the ERA-Interim atmospheric reanalysis (Dee et al., 2011). It includes elevation corrections and seasonal monthly bias corrections from ground-based observations.



## 2.3. Agricultural GY statistics

The Agreste portal (http://agreste.agriculture.gouv.fr/) provides annual statistical surveys over France which allow establishing a database of yearly GY values. The GY estimates are available per crop type and per administrative unit (département). We use GY values for rainfed straw cereals such as barley, oat, rye, triticale and wheat, for the same 45

départements as in Calvet et al. (2012) and Canal et al. (2014). Calvet et al. (2012) and Canal et al. (2014) used Agreste data for the 1994-2008 and 1994-2010 periods, respectively, for both straw cereals and fodder production. We use Agreste data from 1999 to 2013, only for straw cereal GY.

## 3. Methods

### 3.1. The ISBA model

The ISBA LSM is included in the SURFEX (SURFace EXternalisée) modelling platform (Masson et al., 2013). The newest version of SURFEX (version 8) is used in this study with the "NIT" biomass option for ISBA. The "C3 crop" plant functional type is considered.

ISBA simulates the diurnal course of heat, water, and $CO_2$ fluxes. The set of ISBA options we use permits the simulation of LAI and $B_{ag}$ on a daily basis (Calvet et al., 1998, 2008). The model includes a soil moisture stress function (Fs) applied to

photosynthesis key parameters. For low vegetation such as grass or crops, the parameters related to soil moisture stress are (Calvet, 2000): the mesophyll conductance ($g_m$) and the maximum leaf-to-air saturation deficit ($D_{max}$). Values of $g_m$ and $D_{max}$ for straw cereals in well-watered conditions are given in Table 1, together with other model parameters. It must be noted that this value of $g_m$ was derived from IM by Canal et al. (2014) for the same straw cereal sites as those considered in this study. In moderately dry conditions, $g_m$ and $D_{max}$ are affected by Fs in such a way as to increase the intrinsic water use efficiency

(WUE). This corresponds to a drought-avoiding behaviour (Calvet, 2000). The model is also able to represent a drought-tolerant behaviour (stable or decreasing WUE) and Calvet et al. (2012) showed that straw cereals tend to behave as drought-avoiding while grasslands tend to behave as drought-tolerant.

The above-ground biomass ($B_{ag}$) consists of two components within ISBA: the structural biomass and the active biomass. The latter corresponds to the photosynthetically active leaves and is related to $B_{ag}$ by a nitrogen dilution allometric

logarithmic law (Calvet and Soussana, 2001). The mass-based leaf nitrogen concentration ($N_L$) is a parameter of the model affecting the specific leaf area (SLA) which is the ratio of LAI to leaf biomass (in $m^2 kg^{-1}$). The SLA depends on $N_L$ and on plasticity parameters (Gibelin et al., 2006). The $N_L$ parameter is key for LAI simulations and has to be included in any IM experiment involving LAI.

The net assimilation of $CO_2$ by the leaves ($A_n$) is driven by environmental factors such as the atmospheric $CO_2$

concentration, air humidity, the incoming solar radiation and the leaf surface temperature. To upscale the net assimilation of $CO_2$ and transpiration at the vegetation level, a multilayer radiative transfer scheme is used (Carrer et al., 2013). The daily



canopy-scale accumulated value of $A_n$ serves as an input for the vegetation growth and mortality sub-models, and the phenology is completely driven by $A_n$ (no growing degree-day parameterization is used).

The plant transpiration flux is used to calculate the soil water budget through the root water uptake. The soil hydrology scheme used in this study is referred to as "FR-2L" in SURFEX. It represents two soil layers: a thin surface layer with a uniform depth of 1 cm and a root-zone layer of depth Zr. The latter is used as a surrogate for MaxAWC in the calibration process. Soil texture parameters such as the gravimetric fraction of sand and clay are extracted from the Harmonized World Soil Database (Nachtergaele et al., 2012). Physical soil parameters such as volumetric soil moisture at field capacity ($\theta_{Fc}$) and wilting point ($\theta_{Wilt}$) are calculated thanks to pedotransfer functions based on soil texture. The MaxAWC parameter is given by:

$$MaxAWC = \rho(\theta_{Fc} - \theta_{Wilt}) \times Zr \tag{1}$$

Parameters are defined in Table A1 and model parameter values are summarized in Table 1.

### 3.2. Land Data Assimilation System

We used the LDAS described in Barbu et al. (2011, 2014). It consists of a sequential data assimilation system operated offline (uncoupled with the atmosphere). The assimilation is based on a simplified extended Kalman filter (SEKF), able to integrate observations such as LAI and soil moisture in the ISBA model. In this study, only LAI observations are assimilated and the LDAS produces analyzed LAI values.

The key update equation of the SEKF is:

$$\Delta x^t = x_a^t - x_f^t = K(y_o^t - y^t), \text{ with } K = \mathbf{B}\mathbf{H}^{\mathrm{T}}(\mathbf{H}\mathbf{B}\mathbf{H}^{\mathrm{T}} + \mathbf{R})^{-1} \tag{2}$$

where $\Delta x$ is the analysis increment, x is a control vector of one dimension representing LAI values propagated by the ISBA LSM, and $y_o$ is the observation vector representing the GEOV1 LAI observations. The t superscript stands for time (t). The initial time (t = 0) is denoted by the 0 superscript. The "a", "f" and "o" subscripts denote analysis, forecast and observation, respectively. The Kalman gain K is derived from the background error covariance matrix $\mathbf{B}$ and from the observation error covariance matrix $\mathbf{R}$. The $y^t$ term of Eq. (2) represents the model counterpart of the observations, i.e. the model predicted value of the observation at the analysis time:

$$y^t = h(x^0) \tag{3}$$

Matrix $\mathbf{H}$ that appears in Eq. (2) represents the Jacobian of potentially non linear h function:

$$\mathbf{H} = \frac{\partial y^t}{\partial x^0} \tag{4}$$

which gives the following Jacobian vector:



$$\mathbf{H} = \left( \frac{\partial LAI^t}{\partial LAI^0} \right) \qquad (5)$$

The initial state at the beginning of an assimilation window is analysed via the information provided by an observation at the end of the assimilation window (Rüdiger et al., 2010). In this approach, the LAI increments (Eq. 2) are applied at the end of 1-day assimilation intervals. The elements of the Jacobian matrix are estimated by finite differences, individually perturbing each components of the control vector $x$ by a small amount $\delta x$:

$$\mathbf{H} = \frac{y(x + \delta x) - y(x)}{\delta x} \qquad (6)$$

The background error covariance matrix $\mathbf{B}$ is assumed to be constant at the start of each analysis cycle. The covariance matrices $\mathbf{B}$ and $\mathbf{R}$ are assumed to be diagonal. In the simplified version of the EKF used in this study, namely SEKF, the $\mathbf{B}$ matrix does not evolve with time. The standard deviation of errors of GEOV1 LAI is assumed to be 20 % of GEOV1 LAI. The same assumption is made for the standard deviation of errors of the modelled LAI (20 % of modelled LAI) for modelled LAI values higher than 2 $m^2m^{-2}$. For modelled LAI values lower than 2 $m^2m^{-2}$, a constant error of 0.4 $m^2m^{-2}$ is assumed

(following option 3 presented in Barbu et al., 2011).

### 3.3. Upscaling disaggregated LAI observations to département level

Each agricultural spot shown in Fig. 1 corresponds to the area within a département presenting the highest fraction of straw cereals. These 45 locations were chosen by Calvet et al. (2012) on a 8 km × 8 km grid using fractions of vegetation types derived from ECOCLIMAP (Faroux et al., 2013). Disaggregated LAI observations have a spatial resolution of 1 km × 1 km.

This represents a small area compared to the size of a département (from 2000 to 10000 $km^2$). Local values of the straw cereal LAI may not be representative of the straw cereal production at the département level described by Agreste. Preliminary tests showed that averaging the disaggregated LAI on the same 8 km × 8 km grid cell used by Calvet et al. (2012) was not sufficient to represent the interannual variability of the GY observations at the département level. Therefore, an analysis of the consistency of the two observation datasets (in situ GY and disaggregated satellite LAI), is performed. The

average maximum annual value of the disaggregated GEOV1 LAI observation ($LAIo_{max}$) is calculated for various grid cell sizes for this task. In practice, the $LAIo_{max}$ value corresponds to the mean LAI values above a given fraction of the observed maximum annual LAI ($LAI_{max}$). We consider five grid cell sizes of 5 km × 5 km, 15 km × 15 km, 25 km × 25 km, 35 km × 35 km, and 45 km × 45 km (from 25, to 2025 $km^2$). The five $LAIo_{max}$ time series are compared with the GY time series for each département. The area size corresponding to the largest number of départements presenting a significant correlation

between $LAIo_{max}$ and GY is selected.



### 3.4. Model calibration/validation

The feasibility of retrieving MaxAWC from LAI satellite data is explored using two different approaches: IM and LT. For the two approaches, this calibration step is followed by a validation step aiming at demonstrating the relevance the retrieved MaxAWC values and the added value of the retrieval technique.

The satellite LAI observations are available year-round but the sensitivity of straw cereal LAI to MaxAWC may change greatly for one period of the year to another. Prior to calibrating the model, a sensitivity study of the time window used for the MaxAWC retrieval is performed. Three periods are considered: (1) growing period (from 1 March to the date of the observed $LAI_{max}$); (2) peak LAI (period for which observed LAI is higher than 50% of observed $LAI_{max}$) ; (3) senescence (from the date when observed $LAI_{max}$ is reached to 31 July). The ISBA simulations are stopped on 31 July as this date

corresponds to the maximum harvest date at most locations.

The validation of the calibrated model consists in comparing the interannual variability of the simulated maximum annual above-ground biomass to the interannual variability of the GY observations. The 1999-2013 period is considered. In order to limit the impact of model errors, caused for example by uncertainties in the atmospheric forcing, an average value of the simulated $B_{agX}$ is used instead of an instantaneous value. This average value is calculated using all the $B_{ag}$ values above a

threshold corresponding to 90 % of the maximum annual $B_{ag}$. It was checked that this threshold value permits the maximization of the number of départements presenting a significant correlation with GY. Then, scaled anomalies of the average simulated $B_{agX}$ are compared with scaled anomalies of the GY observations, and the $R^2$ score is calculated. Scaled anomalies (As) are calculated using the mean and standard deviation of the two variables over the 1999-2013 period:

$$As_{BagX} = \frac{\left(B_{agX} - \overline{B_{agX}}\right)}{\sigma\left(B_{agX}\right)} \tag{7}$$

$$As_{GY} = \frac{\left(GY - \overline{GY}\right)}{\sigma\left(GY\right)} \tag{8}$$

The interannual variability of the modelled $LAI_{max}$ is assessed using the coefficient of variation (CV). CV is given in % and is calculated according the following formula with $\sigma$ the standard deviation and $\mu$ the mean:

$$CV = 100 \times \sigma / \mu \tag{9}$$

The MaxAWC retrieval is considered to be successful if the Pearson correlation is significant at 1% level (F-test p-value <

25  0.01).



### 3.5. Design of the experiments

#### 3.5.1. Inverse modelling

Two parameters are retrieved: $N_L$ and MaxAWC. For a given value of $N_L$, a set of 13 LAI simulations is produced, corresponding to the following MaxAWC values: 44, 55, 66, 77, 88, 99, 110, 121, 132, 154, 176, 198 and 220 mm. Since $N_L$
is a key parameter for LAI simulations, it has to be retrieved together with MaxAWC and this simulation process is repeated 5 times, for the following $N_L$ values: 1.05, 1.30, 1.55, 2.05 and 2.55 %.

The LAI Root Mean Squared Error (RMSE) is used to select the best simulation by minimising this cost function over the period between the occurrence of the observed $LAI_{max}$ and 31 July for the 15-years:

$$RMSE = \sqrt{\sum_{i=0}^{n} \frac{(LAI_i - LAIo_i)^2}{n}} \tag{10}$$

where LAI is for simulated LAI, LAIo is for observed LAI and $n$ is the length of the data vector.

#### 3.5.2. LDAS tuning

The LAI observations are integrated into ISBA by the LDAS. The LDAS produces analyzed values of LAI and $B_{ag}$. Therefore, there is no need to retrieve $N_L$ and the only degree of freedom in this case is the value of MaxAWC. Thirteen analyses are made, corresponding to the same MaxAWC values used in the IM (Sect. 3.5.1).

The median analysis increment (Eq. 2) can present positive or negative values. Small corrections provided by the LDAS
indicate that simulation outputs are close to observations and that the dynamics is well represented. The value closest to zero indicates the best simulation and the corresponding MaxAWC value is considered as the retrieved MaxAWC.

### 4. Results

#### 4.1. Disaggregated satellite LAI vs. grain yield observations

In a first step before integrating the disaggregated LAI observations into the ISBA model, we checked the consistency of the
interannual variability of $LAIo_{max}$ (Sect. 3.3) with the one of the observed GY from Agreste. We investigated several values of the size of the area around each site coordinates to calculate the average of $LAIo_{max}$, from 25 to 2025 km². Individual $LAIo_{max}$ values at a spatial resolution of 1 km × 1 km correspond to the mean of LAI values above the $LAIo_{max}$ threshold (Sect. 3.3). Several $LAIo_{max}$ threshold values ranging from 40% to 95% of $LAI_{max}$ were investigated together with the grid cell size (see Fig. S1 in Supplement). A $LAIo_{max}$ threshold value of 50 % and a grid cell size of 35 km × 35 km (1225 km²)
were selected. In this configuration, a significant temporal correlation (F-test p-value < 0.01) between the average $LAIo_{max}$ and the observed GY is obtained for 31 départements. The latter are shown in Fig. 1 (empty blue circles). The 45 grid cells of 35 km × 35 km are further used to calculate average 10-day LAI observations to be integrated in the ISBA model through





either IM or LT. The fraction of straw cereals derived from ECOCLIMAP for these grid cells ranges from 15 % to 100 %, with a median value of 68 % (see Table S1 in Supplement).

The temporal correlation between $LAIo_{max}$ and GY is illustrated in Fig. 2. The two 15-year time series correspond to average annual values of $LAIo_{max}$ and GY across the 31 départements where $LAIo_{max}$ is found to correlate with GY. The two time series present a very good correlation, with $R^2 = 0.84$. This shows that the disaggregated satellite-derived LAI is able to capture the interannual variability of GY.

## 4.2. Sensitivity study

Figure 3 presents the impact of ISBA parameters on the simulated annual maximum LAI and on its interannual variability. Two key parameters are considered: MaxAWC and $N_L$. The same parameter values are applied to all 45 départements, and mean modelled $LAI_{max}$ are used to calculate CV values (Eq. 9). The CV values are shown in Fig. 3 as a function of these parameters, together with $LAI_{max}$.

It appears that the interannual variability of the modelled $LAI_{max}$ is governed by MaxAWC. CV values of more than 12 % are derived from the ISBA simulations at low values of MaxAWC (e.g. 50 mm). On the other hand, high MaxAWC values (> 200 mm) correspond to limited interannual variability of $LAI_{max}$ (CV < 4 %), in relation to a lower sensitivity of plants to drought.

The $N_L$ parameter has a limited impact on CV and its impact depends on MaxAWC. For large (small) MaxAWC values above (below) the standard average value of 132 mm used in ISBA, the largest values of $N_L$ tend to cause a decrease (increase) of CV. In the IM experiment, $N_L$ mainly impacts the average simulated $LAI_{max}$ value. In the ISBA model, $N_L$ is linearly related to the leaf specific leaf area (SLA) and large $N_L$ values correspond to large SLA values, i.e. larger LAI values for a given simulated leaf biomass (Gibelin et al., 2006). However, Fig. 3 shows that MaxAWC has a more pronounced impact than $N_L$ on $LAI_{max}$. Increasing MaxAWC from 50 to 250 mm triggers a rise in $LAI_{max}$, from about 2 $m^2m^{-2}$ at low $N_L$ values to 3 $m^2m^{-2}$ at high $N_L$ values. Switching $N_L$ from low to high values at a given MaxAWC level also raises $LAI_{max}$, but not more than 2 $m^2m^{-2}$.

This result confirms that MaxAWC is the key parameter to be retrieved in order to improve the representation of straw cereal biomass, for both IM and LT experiments. The impact of MaxAWC on the cost functions (LAI RMSE and median LAI analysis increments, Eqs. (10) and (2), respectively) may depend on the LAI observation period. We tested the two retrieval methods for three different optimisation periods: start of growing period, peak LAI, and senescence (see Sect. 3.4).

This is illustrated in Fig. 4, which shows the average cost function across all 45 départements. In both experiments, MaxAWC has little impact on the cost function during the start of the growing season. The most pronounced response of both LAI RMSE and analyses increments is observed during the senescence. For this period of the growing cycle, both cost functions present a minimum value at MaxAWC = 110 mm. Also, the largest RMSE and increments values are observed during the senescence, indicating that the processes at stake during this period are more difficult to simulate. For straw cereals, senescence is related to soil moisture stress (Cabelguenne and Debaeke, 1998) and during this period the value of





MaxAWC has a marked impact on the representation of the effect of drought by the model. The peak LAI period is less favourable to the integration of LAI observations into the model, with a reduced accuracy on the retrieved MaxAWC.

### 4.3. Outcomes of the optimisation

A direct result of the optimisation procedure is the reduction of the cost function value. This is illustrated in Fig. 5 for all 45
départements. Figure 5 presents the impact of the optimisation on the cost functions of IM and LT during the senescence period: LAI RMSE and LDAS LAI increments, respectively.

The RMSE values are systematically reduced by the IM. For all 45 départements, the median value of the LAI RMSE drops from 1.6 to 1.2 $m^2m^{-2}$. While LAI RMSE exceeds 1.5 $m^2m^{-2}$ for 29 départements before the optimisation, this RMSE value is exceeded for only three départements after the optimisation. It must be noted that this is a much better result than the RMSE
obtained in Fig. 4 (1.6 $m^2m^{-2}$) for the cost function including all 45 départements, with a MaxAWC value of 110 mm. This shows the impact of the spatial variability of MaxAWC.

For LT, most of the median daily increment values are sharply reduced: while 17 values are larger (smaller) than 0.2 (-0.2) $m^2m^{-2}$ before the optimisation, all the values range from -0.1 to 0.1 $m^2m^{-2}$ after the optimisation. The spatial median value of the LDAS LAI increments varies from -0.03 $m^2m^{-2}$ for original LDAS to -0.01 $m^2m^{-2}$ for LT, for all 45 départements. Table
2 summarizes results showing the impact of the optimization on indicators such as the number of départements presenting a significant correlation of $B_{agX}$ with GY and the median value of the cost functions. Table 2 also give median and standard deviation values of the retrieved MaxAWC and of the retrieved $N_L$ in the case of IM, together with modelled $B_{agX}$ and $LAI_{max}$ values. The results are given for the départements presenting a significant correlation of $B_{agX}$ with GY and for all 45 départements.

In the case of LT, the median retrieved MaxAWC (129±44 mm for all 45 départements and 133±46 mm for significant départements) is close to the standard value used in ISBA (132±2 mm) but the standard deviation is much larger. This shows that LT is able to generate spatial variability in MaxAWC values.

A similar degree of variability is obtained by IM, but the retrieved MaxAWC presents much lower values for all 45 départements: 111±44 mm. On the other hand, a much larger values of 153±40 mm is found for the 16 validated
départements. The retrieved $N_L$ (1.05±0.20) is smaller than the default value of 1.30 %. The role of $N_L$ in the optimization is discussed in Sect. 5.1.

Figure 6 shows the impact of optimizing MaxAWC on the mean annual LAI cycle, with respect to the observed annual LAI cycle over the 45 départements. IM tends to produce a smaller $LAI_{max}$ median value (3.59 $m^2m^{-2}$ for all 45 départements) than basic ISBA simulations or LDAS simulations (3.84 and 3.98 $m^2m^{-2}$, respectively). IM tends to reduce simulated LAI in
May and June, while the LDAS simulations (either original LDAS or LT) are much closer to the observations.





The two optimization methods succeed in reducing the LAI RMSE of the basic ISBA simulations (1.6 $m^2m^{-2}$ for all 45 départements). With optimized MaxAWC, the tuned LDAS annual mean LAI cycle is closer to the observations than LAI resulting from IM, with LAI RMSE equal to 1.1 $m^2m^{-2}$ for LT, against 1.2 $m^2m^{-2}$ for IM.

### 4.4. Validation

The optimisation is considered as successful in départements where the correlation between yearly time series of $B_{agX}$ and GY is significant (p-value < 0.01). Table 2 shows that even without tuning MaxAWC, the integration of LAI in ISBA by the original LDAS permits the increase of the number of départements where p-value < 0.01 from 18 in basic ISBA simulations to 21. LT further increases this number to 24 départements. With only 16 validated départements, IM is not able to outperform original LDAS simulations.

Time series of mean scaled anomalies of $B_{agX}$ and GY are shown in Fig. 7 all 45 départements before and after IM or LT. The marked negative anomalies (< -1) in 2001, 2003 and 2011 are represented well after LT. On the other hand, the impact of sunlight deficit and low temperatures during the growing period of 2001 cannot be represented well after IM. The marked negative GY anomaly observed in 2007 is not very well represented by the model. Moreover, Fig. 7 shows that parameter tuning does not significantly improve $R^2$ values. Basic ISBA and original LDAS simulations present $R^2$ values of 0.65 and
0.80, against 0.65 and 0.82 after IM and LT, respectively.

Figure 8 further shows that the inter-annual variability of $B_{agX}$ is markedly better represented using LT. The scaled modelled $B_{agX}$ and the scaled GY observations averaged over 45 départements present a $R^2$ value of 0.82, against 0.65 for IM. Considering only the successful validated départements, more similar $R^2$ values are observed: 0.88 and 0.80, respectively. Figure 9 presents the spatial correlation between the scaled $B_{agX}$ and the scaled GY observations averaged over the 15-year
period considered in this study. Considering the 45 départements, $R^2$ = 0.61 for LT and $R^2$ = 0.58 for IM. Again, LT supersedes IM, including when the comparison is limited to successfully validated départements, with $R^2$ values of 0.74 and 0.63, respectively.

It must be noted that all the correlations presented in Figs. 8 and 9 are significant, with all p-values smaller than 0.001.

### 4.5. Impact of the optimization technique on MaxAWC retrievals

Differences in validation results can be caused by uncertainties in Agreste GY observations or by the difficulty to upscale the observations and the simulations (Sect. 3.3). In order to limit this effect, we further compared the MaxAWC retrievals and the simulated vegetation variables for a subset of the départements corresponding to the 15 départements which are validated for both IM and LT. Table 2 shows that for this subset of départements, MaxAWC values are similar: 154±40 and 156±40 mm, respectively. On the other hand, vegetation variables are more realistically simulated after LT: median LAI RMSE is
1.2 $m^2m^{-2}$ against 1.4 $m^2m^{-2}$ for IM. The median $LAI_{max}$ value is much larger for LT: 4.35 $m^2m^{-2}$, against 3.85 $m^2m^{-2}$ for IM. However, peak $B_{ag}$ values are similar: 1.26 kg $m^{-2}$ for LT and 1.23 kg $m^{-2}$ for IM.



The similarity in MaxAWC retrievals and the contrasting simulated LAI values are illustrated in Fig. 10. Analyzed LAI from LT is closer to the LAI observations than the simulated LAI resulting from IM. The MaxAWC retrievals are slightly smaller for IM and correlate very well with the MaxAWC retrievals from LT ($R^2 = 0.81$). The latter result is also valid when all 45 départements are considered, with $R^2 = 0.72$.

**5. Discussion**

**5.1. What is the added value of the LDAS ?**

The LDAS approach allows sequential integration of LAI observations into the model. Minimizing analysis increments to retrieve MaxAWC is a much more complex approach than IM. Overall, MaxAWC retrievals from the two methods are relatively  consistent (see Sect. 4.5) but IM tends to produce smaller values. On the other hand, GY observations show that
the simulated vegetation variables are more realistically simulated after LT than after IM. The LAI simulations are more realistic and $B_{ag}$ simulations are also more realistic (see Figs. 7-10). This can be explained by a better capability of the LDAS to use the observations to drive the model trajectory: the sequential assimilation of LAI is able to constrain the simulated LAI values.

Another advantage of LT is that $N_L$ does not have to be determined, because LAI is directly constrained by the LAI
observations. It can be shown that the impact of tuning $N_L$ in the IM method can be significant. Table 3 presents MaxAWC and LAI RMSE values obtained from IM when only one parameter, MaxAWC, is optimized. Results are shown for five values of $N_L$ ranging from 1.05 % to 2.55 %. The number of validated départements drops when $N_L$ increases, from 16 at $N_L$ = 1.05 % to only 3 at $N_L$ = 2.55 %. At the same time, the MaxAWC retrievals tend to present smaller values, down to 88±40 mm at $N_L$ = 2.55 %. This result can be explained by the fact that larger values of either MaxAWC or $N_L$ tend to increase
$LAI_{max}$ (Fig. 3).

Improving the simulation of vegetation variables has a positive impact on the quality of simulated hydrological variables such as evapotranspiration and soil moisture (Szczypta et al., 2012). Therefore, the larger MaxAWC values obtained from LT (129±44 mm) are likely to be more realistic than those obtained from IM (111±44 mm).

**5.2. Are MaxAWC retrievals and simulated peak $B_{ag}$ realistic ?**

In order to verify the MaxAWC values derived from LAI observations, we extracted MaxAWC values from a map produced by Institut National de la Recherche Agronomique (INRA) a spatial resolution of 1 km × 1 km. This map was established using pedotransfer functions based on soil physical properties information such as soil texture, soil depth, bulk density, and organic matter (Al Majou et al., 2008). A given local MaxAWC value corresponds to a soil typological unit (STU). The 1 km × 1 km soil mapping units may contain several STUs and the STU fraction is known. We computed weighted-average
MaxAWC values for every 35 × 35 km grid cell. The resulting INRA MaxAWC values of the 45 départements present a



median value of 151 mm and a standard deviation of 54 mm. This confirms that the MaxAWC values obtained from LT (129±44 mm) are more realistic than those obtained from IM (111±44 mm).

The median peak $B_{ag}$ values are about 1.2 kg m$^{-2}$ in all simulations. This is consistent with total maximum above-ground biomass values for cereals, which range between 1.1 and 1.7 kg m$^{-2}$ (e.g. Loubet et al., 2011). Because $B_{agX}$ corresponds to

the mean $B_{ag}$ above 90 % of the peak $B_{ag}$ value (Sect. 3.4), median $B_{agX}$ values are smaller than peak $B_{ag}$ and do not exceed 1 kg m$^{-2}$ (Table 2).

### 5.3. Are LAI satellite data suitable for the optimisation of MaxAWC ?

The optimization methods used in this study are based on disaggregated LAI satellite data and the quality of the results depends on the reliability of the observation dataset. The MaxAWC parameter is a crucial parameter for the senescence

period, between $LAI_{max}$ and harvesting (Fig. 4). Because $LAI_{max}$ is related to a large extent to MaxAWC (Fig. 3d), an underestimation of observed maximum LAI values would force the retrieval method to underestimate MaxAWC.

From this point of view, using disaggregated LAI observations is key. Figure 11 compares the mean of annual maximum values of raw LAI and disaggregated LAI for the 45 départements and for 1225 km$^2$ grid cells. Using disaggregated LAI increases the observed value of maximum LAI by up to 40% with respect to raw LAI. The mean difference is 0.43 m$^2$m$^{-2}$.

This mitigates a marked underestimation of the MaxAWC retrievals. As shown in Table 2, the MaxAWC values obtained from LT (110±38 mm) and from IM (83±30 mm) are much lower (15 to 25 %) than those retrieved using disaggregated LAI observations. Moreover, the number of validated départements using GY observations presenting significant positive correlation is reduced: only 10 and 18 for IM and LT, against 16 and 24 with disaggregated LAI, respectively. Also, peak $B_{ag}$ values (for all 45 départements) are smaller: 1.01 and 1.08 kg m$^{-2}$, against 1.14 and 1.17 kg m$^{-2}$ with disaggregated LAI,

respectively.

### 5.4. Can model simulations predict the relative gain or loss of agricultural production during extreme years ?

The observed disaggregated LAI and GY in Fig. 2 show that 2004, 2008, 2009, and 2012 were favourable years for straw cereal production, while 2001, 2003, 2007, and 2011 were unfavourable years. Unfavourable conditions for straw cereal production were caused by droughts, by excess of water, or by a deficit in solar radiation. For example, the 2000-2001 winter

was characterized by extensive floods and by a deficit of solar irradiance until the end of the spring. These climate events markedly affected plant growth especially in northern France (Agreste Bilan, 2001). The 2003 and 2011 years were particularly warm, with a marked precipitation deficit at springtime (Agreste Bilan, 2003, 2011)). Concerning 2007, although climate conditions were favourable to plant growth during spring, extremely wet conditions occurred at the end of the growing season. This triggered accessibility issues and disease development (Agreste Bilan, 2007). These processes limiting

biomass production in response to an excess of water are not represented in the ISBA model. However, the continuous constraint on the model applied by the LDAS on simulated vegetation variables allows the indirect representation of these





adverse effects. This is illustrated in Fig. 7: the negative anomaly of 2007 is much better represented by the LDAS than by simple ISBA simulations.

### 5.6. Can observed LAI characteristics be used to retrieve MaxAWC ?

We investigated the use of a simple statistical analysis of the disaggregated LAI observations to retrieve MaxAWC. Figure 3 shows that there is a marked relationship between MaxAWC and the simulated $LAI_{max}$ and LAI CV. To what extent are these relationships observable ?

In order to answer this question, we used the LT MaxAWC retrievals as a reference dataset. We compared the observed median annual maximum LAI and LAI CV with MaxAWC. No significant correlation could be shown for LAI CV, with $R^2$ smaller than 0.2. On the other hand, a very good correlation ($R^2 = 0.70$ for all 45 départements) was found for median annual maximum LAI (Fig. 12). Using this simple linear regression model, MaxAWC can be estimated with a RMSE of 28.7 mm. A very similar result is obtained considering only the 24 validated départements for LT. This shows that satellite-derived LAI observations have potential to map MaxAWC very simply. The modelled MaxAWC values are given in Table S1 (see Supplement).

### 6. Conclusion

Satellite data are used to optimize a key parameter of the ISBA land surface model for straw cereals in France: the maximum available soil water content, MaxAWC. Two optimization methods are used. IM consists in minimizing the LAI RMSE and LT consists in minimizing LAI analyses increments. The added value of the optimization is evaluated using simulated above-ground biomass, through its correlation with in situ grain yield observations.

It is found that disaggregated LAI observations during the senescence are more informative than raw LAI observations and than LAI observations during the growing phase. The best results are obtained using LT: the simulated above-ground biomass correlates better with gain yields observations, and the retrieved MaxAWC values are more realistic. It is shown that LDAS simulations can predict the relative gain or loss of agricultural production during extreme years, much better than model simulations even after parameter optimization.

Finally, it is shown that median annual maximum disaggregated LAI observations correlate with MaxAWC retrievals over France. This simple metric derived from LAI observations could be used to map MaxAWC. More research is needed to investigate to what extent this conclusion holds for other regions of the world and other vegetation types.





## Appendix A

**Table A1**: Nomenclature.

| List of symbols | |
|---|---|
| $A_{S,BagX}$ | Scaled anomaly of $B_{agX}$ of a given year (-) |
| $A_{S,GY}$ | Scaled anomaly of GY of a given year (z score) (-) |
| AWC | Simulated Available soil Water Content (kg m$^{-2}$) |
| $B_{ag}$ | Simulated living above-ground biomass (kg of dry matter m$^{-2}$) |
| $B_{agX}$ | Maximum of simulated living above-ground biomass (kg of dry matter m$^{-2}$) |
| CV | Coefficient of Variation (%) |
| $D_{max}$ | Maximum leaf-to-air saturation deficit (kg kg$^{-1}$) |
| Fs | Soil moisture stress function |
| $g_m$ | Mesophyll conductance in well-watered conditions (mm s$^{-1}$) |
| GY | Annual Grain Yields of crops (kg m$^{-2}$) |
| IM | Inverse modelling |
| LAI | Leaf Area Index (m$^2$ m$^{-2}$) |
| LDAS | Land Data Assimilation System |
| LSM | Land Surface Model |
| LT | LDAS tuning |
| MaxAWC | Maximum Available soil Water Content (mm or kg m$^{-2}$) |
| NIT | Photosynthesis-driven plant growth version of ISBA-A-gs |
| $N_L$ | Leaf nitrogen concentration (% of leaf dry mass) |
| SLA | Specific Leaf Area (m$^2$ kg$^{-1}$) |
| WUE | Leaf level Water Use Efficiency (ratio of net assimilation of $CO_2$ to leaf transpiration) |
| Zr | Depth of the root zone layer (m) |
| **Greek symbols** | |
| $\rho$ | Water density (kg m$^{-3}$) |
| $\theta$ | Volumetric soil water content (m$^3$ m$^{-3}$) |
| $\theta_{Fc}$ | Volumetric soil water content at field capacity (m$^3$ m$^{-3}$) |
| $\theta_{Wilt}$ | Volumetric soil water content at wilting point (m$^3$ m$^{-3}$) |



*Acknowledgments.* The work of Hélène Dewaele was supported by CNES and by Météo-France. The work of Simon Munier was supported by European Union Seventh Framework Programme (FP7/2007-2013) under grant agreement no. 603608, "Global Earth Observation for integrated water resource assessment" (eartH2Observe). The work of Nabil Laanaia was supported by the Belgian MASC BELSPO project under contract no. BR/121/A2/MASC. The authors would like to thank

the Copernicus Global Land service for providing the satellite-derived LAI products.

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



**Table 1**. Default ISBA parameter values for straw cereals ("C3 crops") in SURFEX V8 for the considered 45 départements.

| Parameter name | Symbol | Value | Units | Reference |
|---|---|---|---|---|
| Rooting depth | $Z_r$ | 1.5 | m | |
| Soil moisture at wilting point | $\theta_{Wilt}$ | 0.12 to 0.28 | $m^3\ m^{-3}$ | |
| Soil moisture at field capacity | $\theta_{Fc}$ | 0.20 to 0.37 | $m^3\ m^{-3}$ | |
| Soil moisture at saturation | $\theta_{Wsat}$ | 0.42 to 0.48 | $m^3\ m^{-3}$ | |
| Behaviour in dry conditions | | drought-avoiding | | Calvet et al. (2012) |
| Leaf nitrogen concentration (mass-based) | $N_L$ | 1.3 | % of dry matter mass | Gibelin et al. (2006) |
| Maximum air saturation deficit | $D_{max}$ | 0.05 | $kg\ kg^{-1}$ | Gibelin et al. (2006) |
| Mesophyll conductance | $g_m$ | 1.75 | $mm\ s^{-1}$ | Canal et al. (2014) |
| Cuticular conductance | $g_c$ | 0.25 | $mm\ s^{-1}$ | Gibelin et al. (2006) |
| Minimum LAI value | $LAI_{min}$ | 0.3 | $m^2\ m^{-2}$ | Gibelin et al. (2006) |

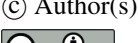



**Table 2**. Impact of the optimization (either inverse modelling -IM- or LDAS tuning -LT-) on parameter values (spatial median values ± standard deviation) of the ISBA model (MaxAWC and $N_L$), on the median value of $B_{agX}$ and $LAI_{max}$, on peak simulated $B_{ag}$, and on the models scores during the senescence period with respect to the disaggregated LAI observations. The results are given for the validated départements, i.e. those presenting a significant correlation (p-value < 0.01) of $B_{agX}$ with Agreste straw cereal grain yield observations. Results for all 45 départements are given in brackets and in italics. The * symbol indicates results obtained using raw LAI observations (undisaggregated). Results for 15 validated départements for both inverse modelling or LDAS tuning are in square brackets. Parameter values resulting from the optimization are in bold. Because simulated $LAI_{max}$ and $B_{agX}$ vary from one year to another, spatial median values are based on median temporal values across the considered 15 year period.

| | Basic ISBA | IM | Original LDAS | LT |
|---|---|---|---|---|
| Number of départements presenting significant positive correlations (p-value < 0.01) | 18 <br> 9* | 16 <br> 10* | 21 <br> 18* | 24 <br> 18* |
| MaxAWC (mm) | 132 ± 2 <br> *(132 ± 2)* | **153 ± 40** <br> *(111 ± 44)* <br> **113 ± 40*** <br> *(83 ± 30)** <br> **[154 ± 40]** | 132 ± 2 <br> *(132 ± 2)* | **133 ± 46** <br> *(129 ± 44)* <br> **106 ± 42*** <br> *(110 ± 38)** <br> **[156 ± 40]** |
| $N_L$ (%) | 1.30 <br> (constant value) | **1.05 ± 0.20** <br> *(1.05 ± 0.20)* <br> *(1.05 ± 0.17)** <br> **[1.05 ± 0.20]** | 1.30 <br> (constant value) | 1.30 <br> (constant value) |
| $B_{agX}$ (kg m$^{-2}$) | 0.99 ± 0.05 <br> *(1.01 ± 0.07)* <br> 0.99 ± 0.03* <br> *(1.01 ± 0.07)** | 0.96 ± 0.16 <br> *(0.89 ± 0.16)* <br> 0.74 ± 0.15* <br> *(0.75 ± 0.11)** <br> [0.98 ± 0.16] | 0.96 ± 0.07 <br> *(0.93 ± 0.11)* <br> 0.88 ± 0.10* <br> *(0.88 ± 0.13)** | 0.98 ± 0.17 <br> *(0.97 ± 0.17)* <br> 0.74 ± 0.17* <br> *(0.84 ± 0.17)** <br> [1.04 ± 0.14] |
| Peak $B_{ag}$ (kg m$^{-2}$) | 1.20 ± 0.05 <br> *(1.22 ± 0.07)* <br> 1.22 ± 0.05* <br> *(1.22 ± 0.07)** | 1.18 ± 0.09 <br> *(1.14 ± 0.13)* <br> 1.01 ± 0.13* <br> *(1.01 ± 0.11)** <br> [1.23 ± 0.08] | 1.20 ± 0.10 <br> *(1.17 ± 0.14)* <br> 1.12 ± 0.12* <br> *(1.12 ± 0.16)** | 1.19 ± 0.18 <br> *(1.17 ± 0.18)* <br> 1.01 ± 0.22* <br> *(1.08 ± 0.19)** <br> [1.26 ± 0.12] |





| | | | | |
|---|---|---|---|---|
| LAI$_{max}$ (m$^2$m$^{-2}$) | $3.84 \pm 0.29$ | $3.83 \pm 0.47$ | $4.17 \pm 0.26$ | $4.15 \pm 0.53$ |
| | *(3.84 ± 0.30)* | *(3.59 ± 0.46)* | *(3.98 ± 0.3)* | *(3.95 ± 0.52)* |
| | $3.52 \pm 0.45*$ | $3.67 \pm 0.37*$ | $3.91 \pm 0.35*$ | $3.51 \pm 0.61*$ |
| | *(3.73 ± 0.38)\** | *(3.42 ± 0.40)\** | *(3.99 ± 0.39)\** | *(3.81 ± 0.55)\** |
| | | [3.85 ± 0.45] | | [4.35 ± *0.40*] |
| LAI RMSE (m$^2$m$^{-2}$) | $1.6 \pm 0.1$ | $1.4 \pm 0.2$ | $1.2 \pm 0.1$ | $1.1 \pm 0.2$ |
| | *(1.6 ± 0.2)* | *(1.2 ± 0.2)* | *(1.3 ± 0.1)* | *(1.1 ± 0.1)* |
| | $1.8 \pm 0.3*$ | $1.2 \pm 0.2*$ | $1.2 \pm 0.1*$ | $1.0 \pm 0.1*$ |
| | *(1.7 ± 0.3)\** | *(1.2 ± 0.2)\** | *(1.2 ± 0.1)\** | *(1.1 ± 0.1)\** |
| | | [1.4 ± 0.2] | | [1.2 ± 0.1] |
| Median LAI increments (m$^2$m$^{-2}$) | | | $0.06 \pm 0.28$ | $-0.01 \pm 0.03$ |
| | | | *(-0.03 ± 0.33)* | *(-0.01 ± 0.03)* |
| | | | $-0.21 \pm 0.33*$ | $-0.01 \pm 0.12*$ |
| | | | *(-0.21 ± 0.33)\** | *(-0.01 ± 0.08)\** |
| | | | | [-0.01 ± 0.03] |





**Table 3**. Impact of $N_L$ on MaxAWC retrieval using a single-parameter inverse modelling technique. The retrieved median MaxAWC and LAI RMSE are given for all 45 départements together with their standard deviation.

| $N_L$ (%) | 1.05 | 1.30 | 1.55 | 2.05 | 2.55 |
|---|---|---|---|---|---|
| Number of départements presenting significant positive correlations (p-value < 0.01) | 16 | 13 | 12 | 6 | 3 |
| MaxAWC (mm) | 110 ± 44 | 110 ± 44 | 99 ± 43 | 99 ± 41 | 88 ± 40 |
| LAI RMSE ($m^2m^{-2}$) | 1.2 ± 0.2 | 1.2 ± 0.2 | 1.3 ± 0.2 | 1.4 ± 0.2 | 1.5 ± 0.2 |





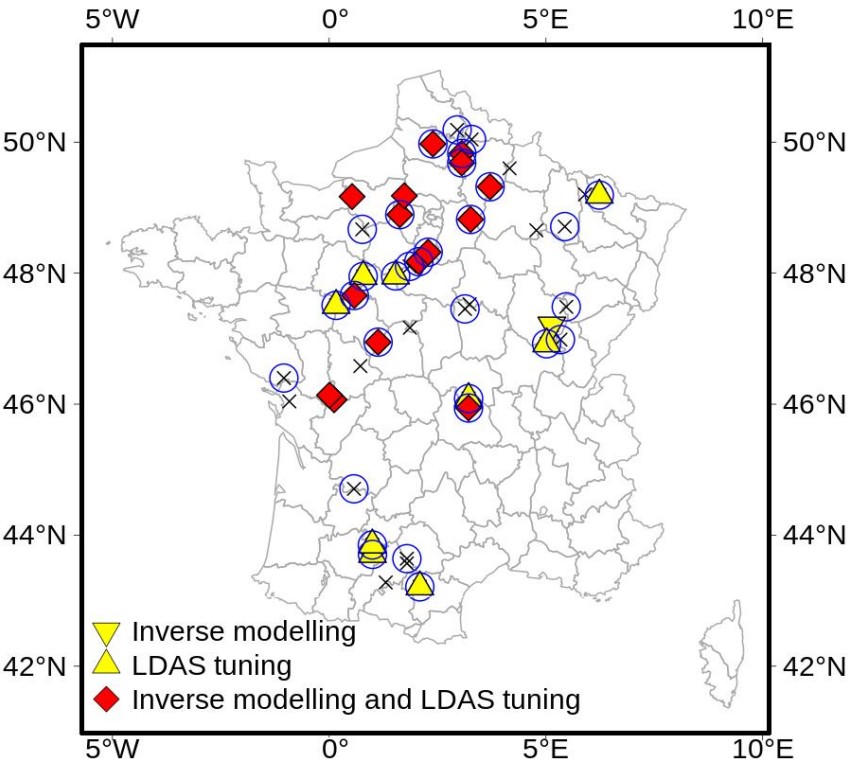

**Figure 1. Straw cereal sites (35 km × 35 km) in France in 45 administrative units ("départements"). Colour symbols show the départements presenting a significant temporal correlation ($R^2 > 0.41$, F-test p-value < 0.01) between Agreste GY values and (empty blue circles) LAIo$_{max}$, (red diamonds) both inverse modelling and LDAS tuning, (yellow up triangle) LDAS tuning only, (yellow down triangle) inverse modelling only. The "×" symbol indicates départements where no significant correlation between biomass simulations and GY could be found.**





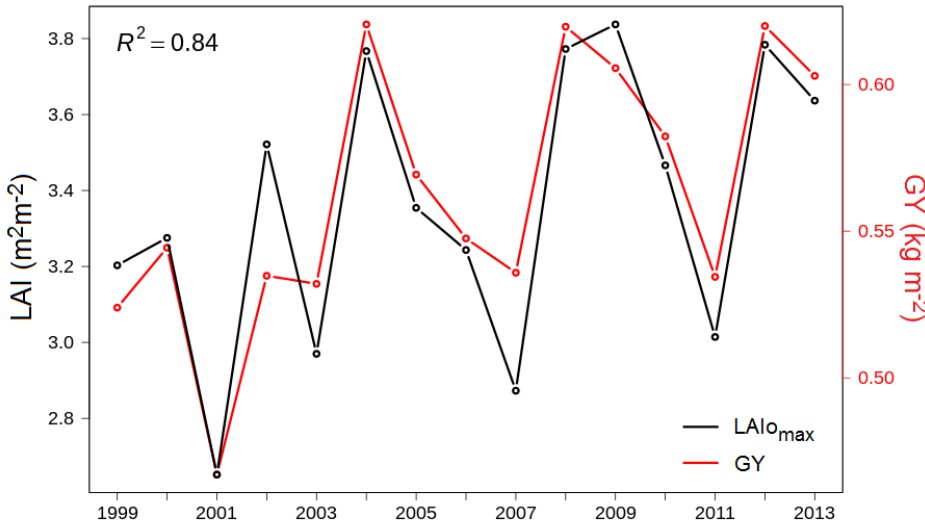

**Figure 2. Interannual variability of straw cereals in France: fifteen-year time series (1999-2013) of the mean disaggregated satellite-derived $LAIo_{max}$ and of the mean Agreste grain yield (GY) observations for 31 French départements where $LAIo_{max}$ and GY are significantly correlated. The fraction of explained variance by the mean $LAIo_{max}$ is $R^2 = 0.84$.**





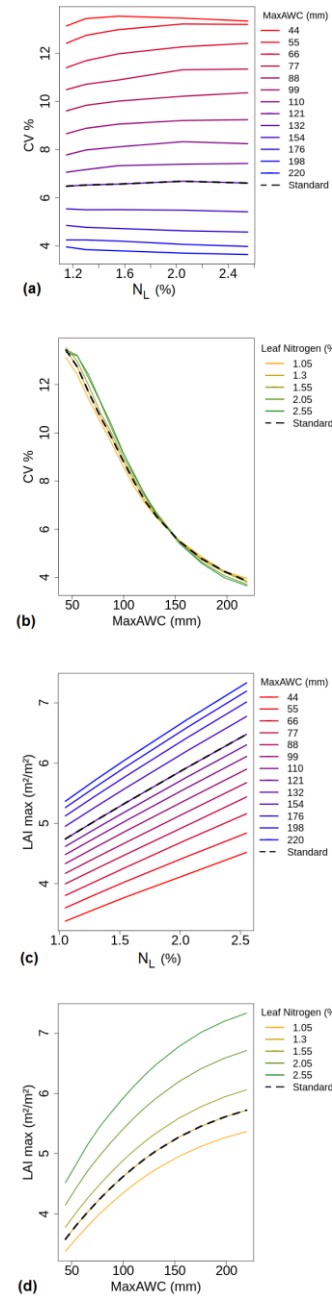

**Figure 3. Impact of MaxAWC and $N_L$ parameters on the annual maximum LAI simulated by ISBA and on its interannual variability. The interannual variability is quantified using the coefficient of variation (CV, in %). Mean CV values (across all 45 départements) are plotted as a function of (a) $N_L$, and (b) MaxAWC, for various values of MaxAWC and $N_L$, respectively. Mean LAI$_{max}$ values (across all 45 départements) are plotted as a function of (c) $N_L$, and (d) MaxAWC, for various values of MaxAWC and $N_L$, respectively. The dashed lines are obtained using standard average ISBA parameter values (MaxAWC = 132 mm and $N_L$ = 1.3 %).**




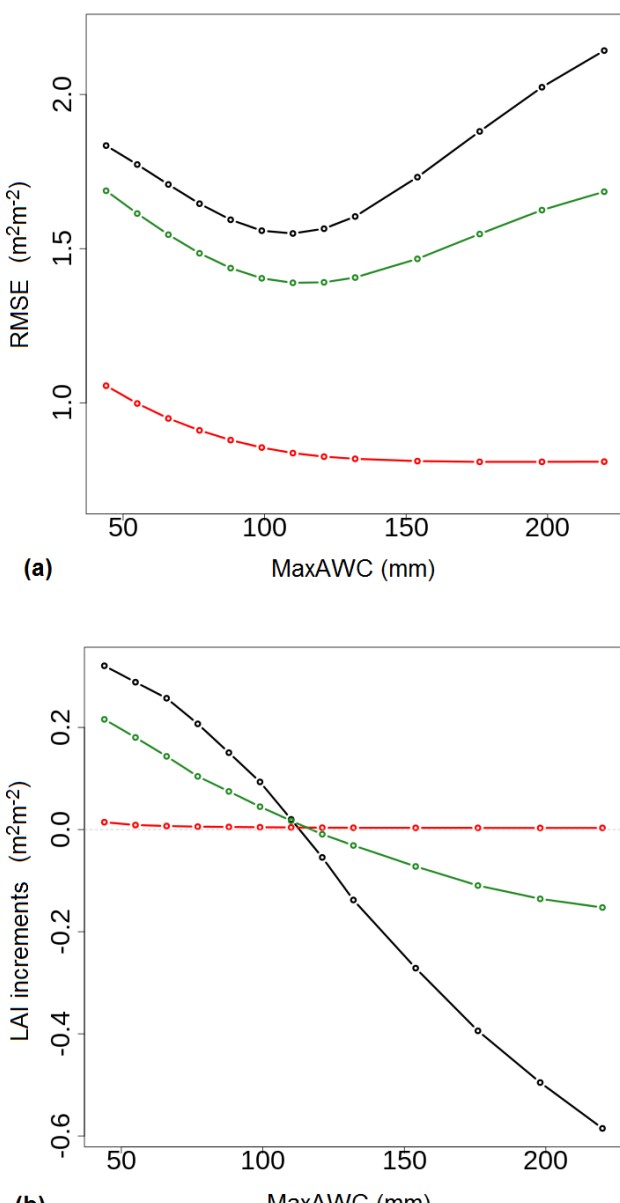

**Figure 4. Mean cost function values vs. MaxAWC across all 45 départements of (a) IM and (b) LT experiments for three different optimisation periods: (red) start of growing period (from 1 March to $LAI_{max}$ date), (green) peak LAI (dates for which LAI > 0.5 $LAI_{max}$), (dark) senescence (from LAImax date to 31 July). IM is based on the minimization of LAI RMSE (Eq. 10). LT is based on the minimization of the median LAI analysis increment (Eq. 2).**




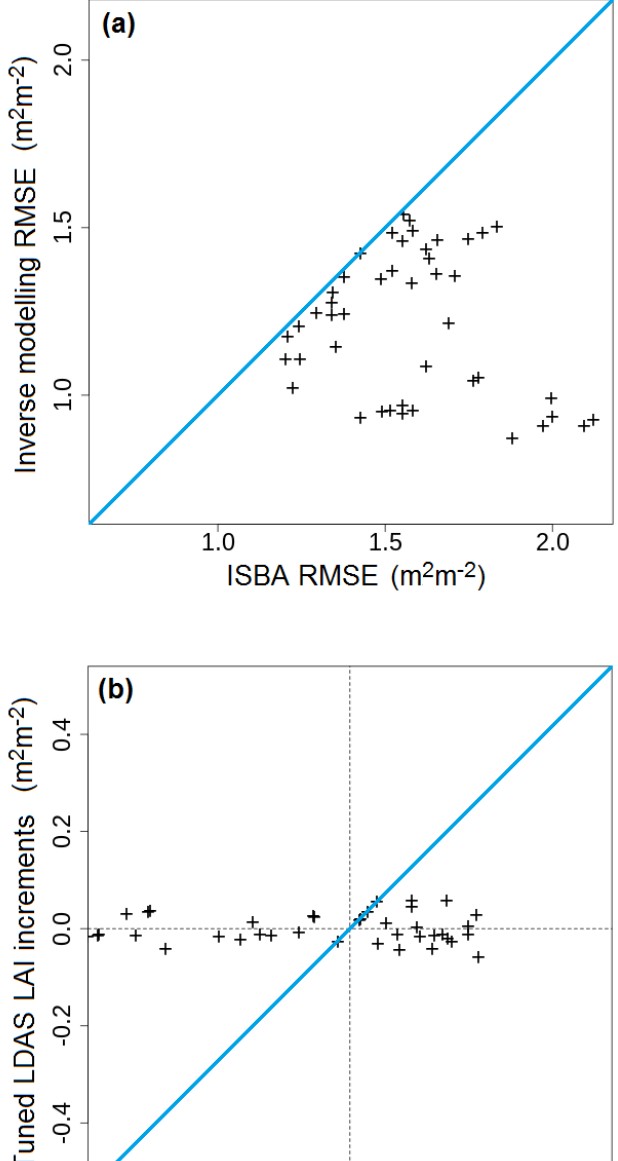

**Figure 5. Cost function values during the senescence period after vs. before LAI observation integration for all 45 départements: (a) LAI RMSE (Eq. 10) for IM, (b) LAI analysis increments (Eq. 2) for LT. Identity lines are in blue.**





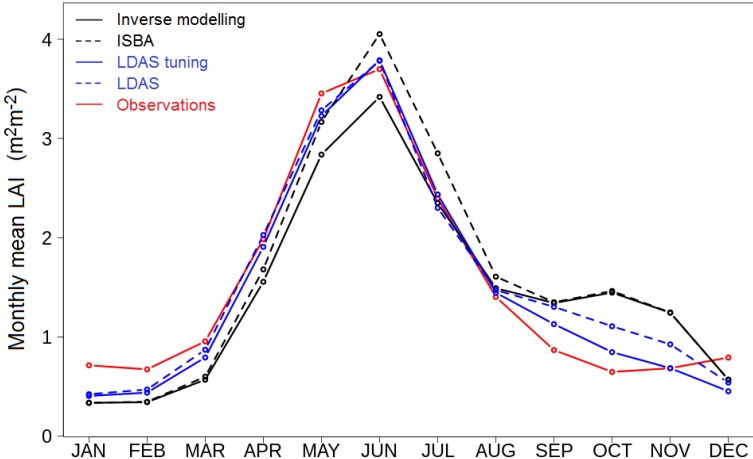

**Figure 6. Mean LAI annual cycle of straw cereals over France (45 départements) during the 1999-2013 period: (red line) satellite-derived observations, (dark dashed line) basic ISBA simulation, (blue dashed line) original LDAS simulation, (solid dark line) IM simulation, (solid blue line) LT simulation.**

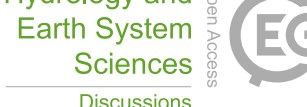

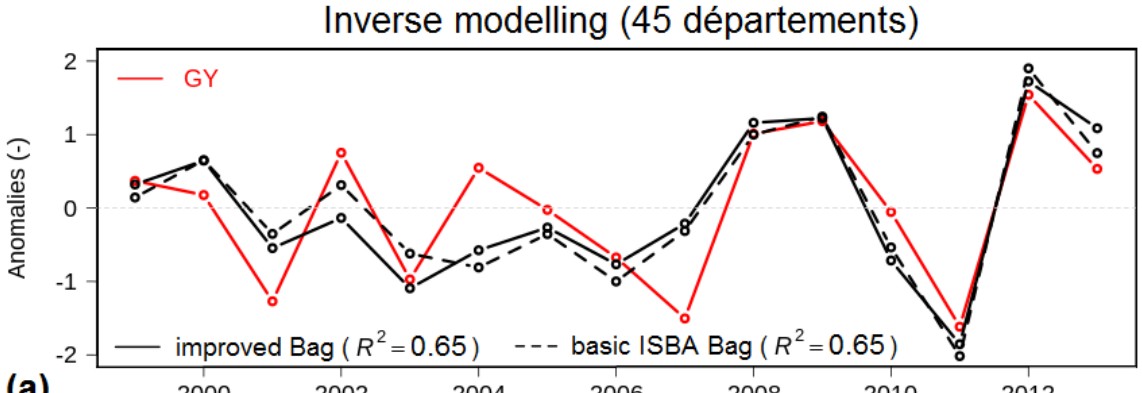

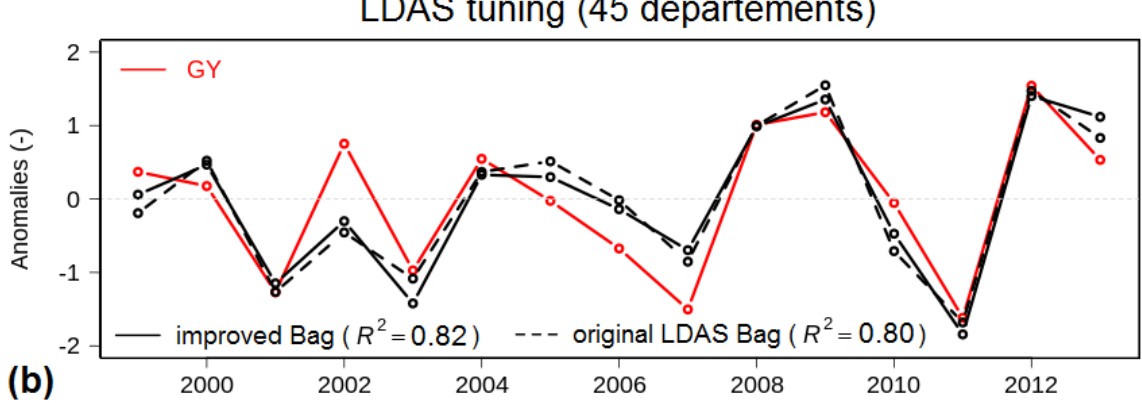

**Figure 7. Scaled GY observation anomalies (As$_{GY}$) and scaled simulated B$_{agX}$ anomalies (As$_{BagX}$) after LAI observation integration for all 45 départements: (a) IM, (b) LT. Red lines are for observations, dark lines are for simulations, dark dashed line is for the original un-tuned simulations. The fraction of explained variance by As$_{BagX}$ is R$^2$ = 0.65 for IM, and 0.82 for LT.**





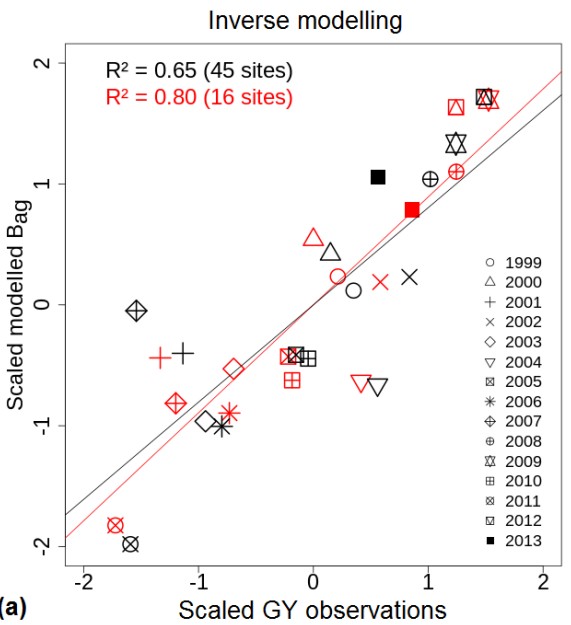

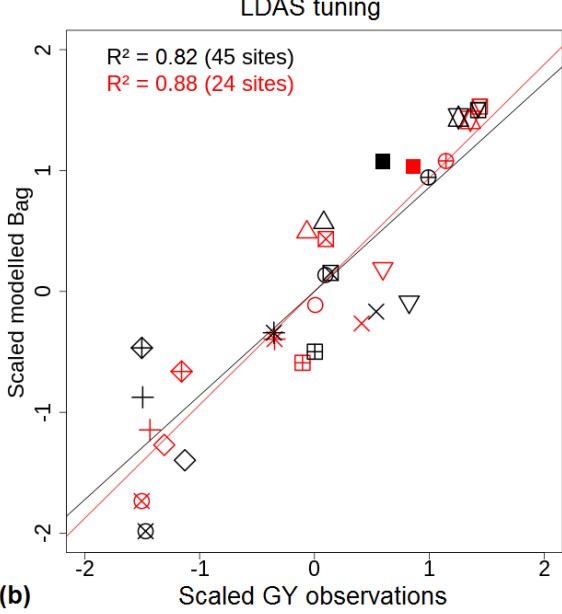

**Figure 8. Temporal correlation between As$_{BagX}$ and As$_{GY}$ for (red symbols) départements presenting significant positive correlations (p-value < 0.01) between simulated B$_{agX}$ and GY and (red and dark symbols) all 45 départements, and for (a) IM, (b) LT.**




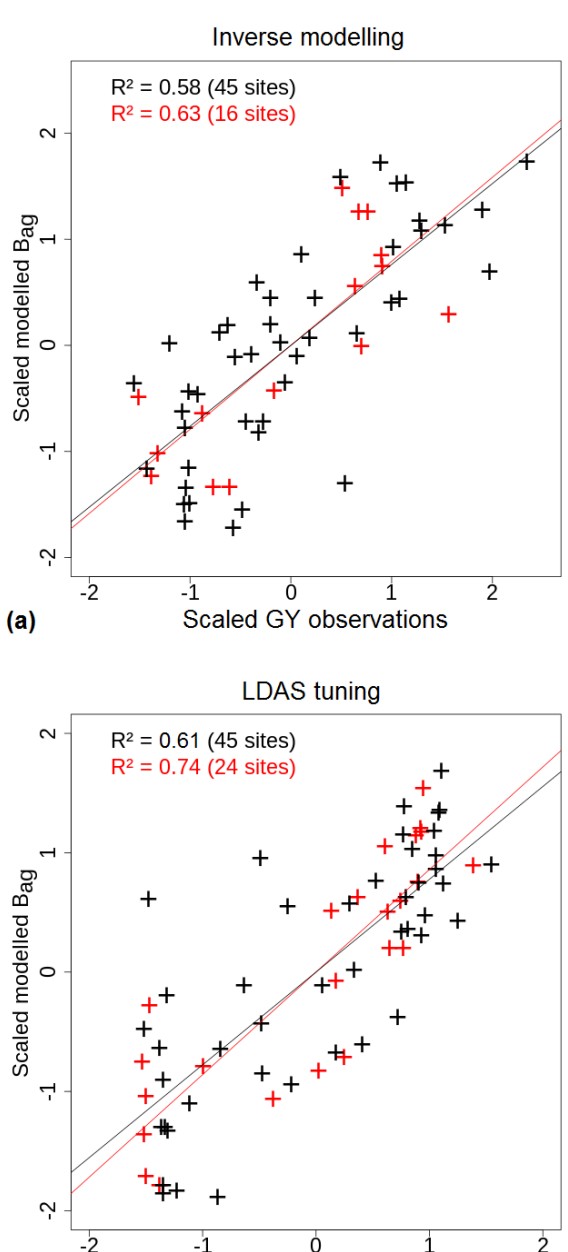

**Figure 9. As in Fig. 8, except for spatial correlation.**





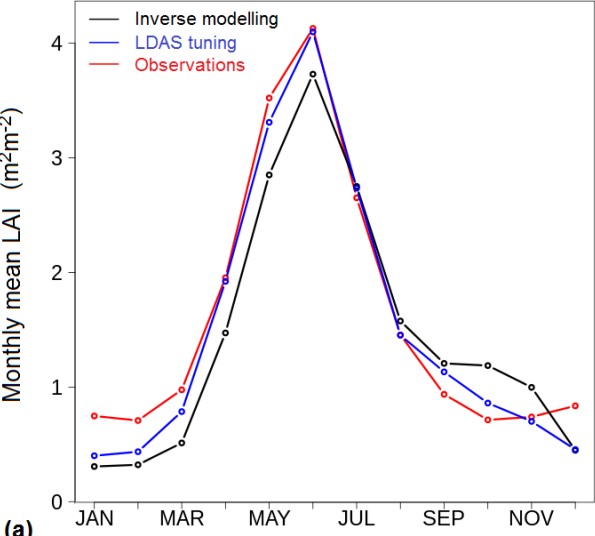

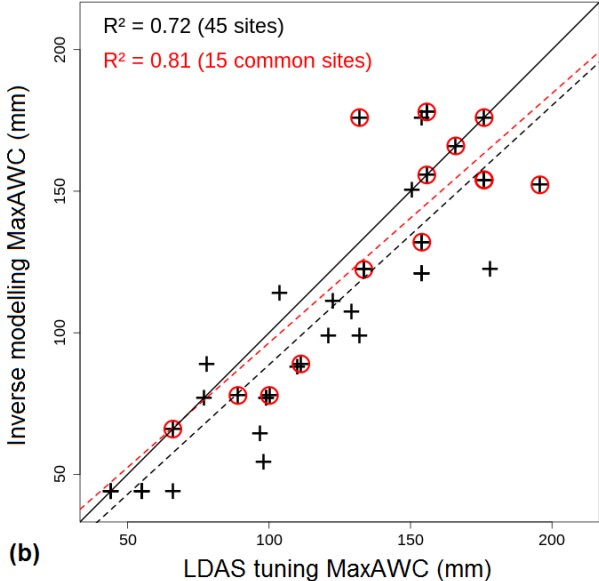

**Figure 10. IM vs. LT: (top) mean LAI annual cycle for 15 validated départements with both methods from (red line) satellite-derived observations, (dark line) IM, (blue line) LT; (bottom) MaxAWC comparison for (+) all 45 départements (R2 = 0.72) and for (red circles) the 15 common départements (R$^2$ = 0.81).**




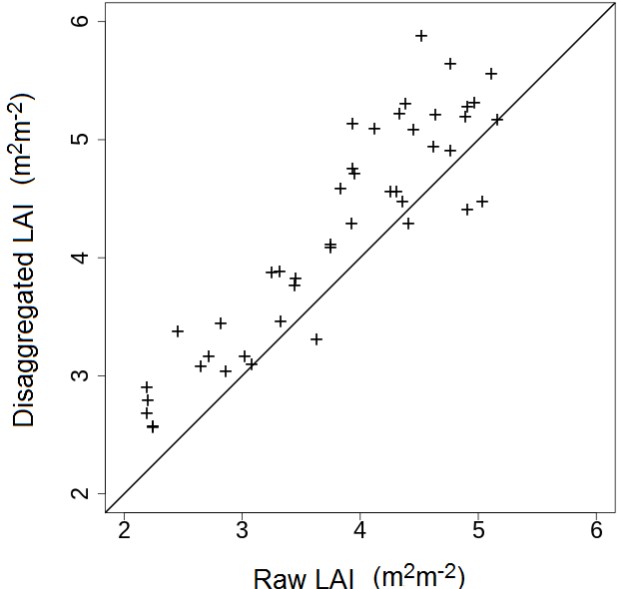

**Figure 11. Comparison of mean annual LAI$_{max}$ of the raw GEOV1 product and of the disaggregated GEOV1 product.**




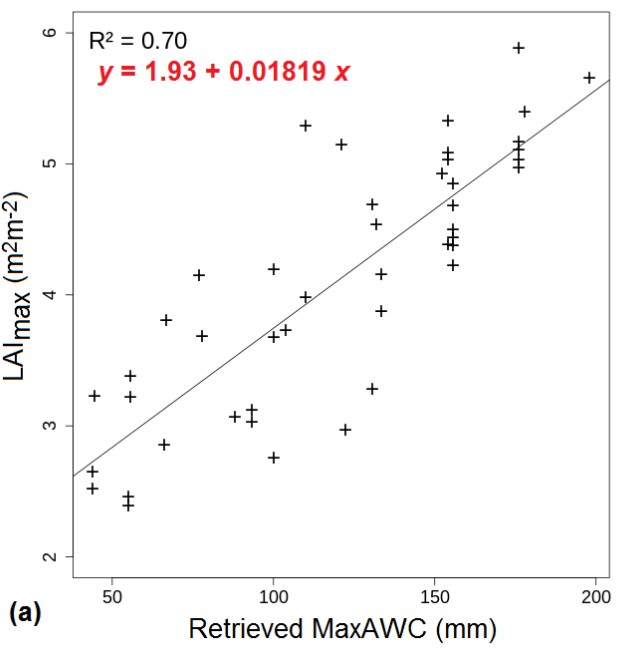

**(a)**

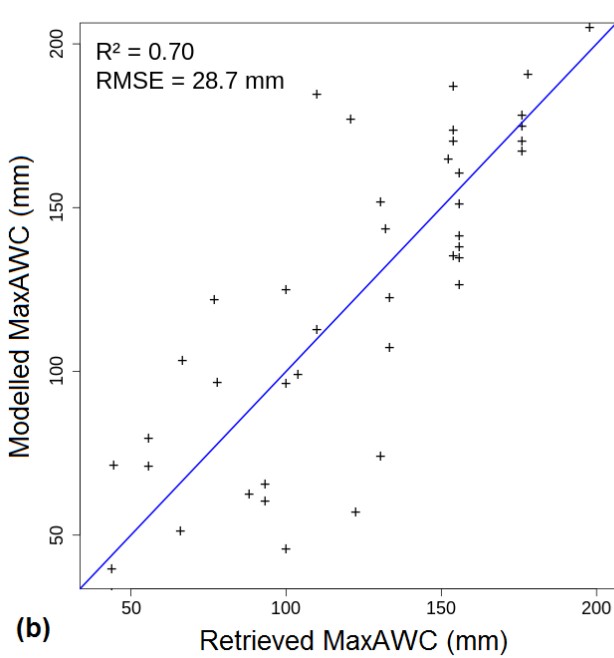

**(b)**

**Figure 12. Use of median observed annual maximum of LAI to retrieve MaxAWC for all 45 départements: (a) linear regression relationship between maximum LAI and the LT MaxAWC retrievals, and (b) MaxAWC estimates derived from the statistical model using maximum LAI observations as a predictor vs. the LDAS tuning MaxAWC retrievals.**