# Peer review of "Parameter optimisation for a better representation of drought by LSMs: inverse modelling vs. sequential data assimilation"

_Hydrology and Earth System Sciences, 2017_

## Referee Comment (RC1) · Anonymous Referee #1 · 25 Apr 2017

Paper summary

This paper describes work done to estimate an uncertain land surface parameter (Max-AWC). This parameter is particularly relevant for land-atmosphere coupling (e.g. transpiration) and for agricultural simulation. Improved knowledge of the parameter is important and will have implications for climate change impact studies as well as shorter term seasonal forecasting.

The authors have used satellite observations of leaf area index (LAI) along with a soil/biosphere/atmosphere model to optimally estimate MaxAWC. The two methods they use are a simple inverse modelling method (essentially choosing the parameter which minimises the RMSE between simulated and observed biomass production),

and a more sophisticated method which makes use of the data assimilation capability of the model in order to estimate the parameter. The results show that the more sophisticated method gives better estimation of the parameter. Interestingly, they also find that optimal MaxAWC is well correlated with the satellite LAI, suggesting that LAI could be used as a simple method of estimating MaxAWC. However as the authors state, caution should be exercised in extrapolating these results beyond the focus of the study, without further research.

General comments

The work is technically sound, scientifically interesting and worthy of publication. However I do suggest some revisions to the text for clarity and readability and beyond these specific revisions recommend further proof reading by the authors, a native English speaker and/or the journals editorial team. Particular attention should be paid to clarity in the introduction as improvements here would encourage more readers to engage with the paper.

Specific comments

P1 L8 "this parameter is usually unavailable" - slightly awkward, perhaps "this parameter is uncertain"

P1 L23 "supervision" - not sure what is meant by this

P1 L29 "This quantity..." - this sentence is confusing and could be improved; please bear in mind any non-expert readers (e.g. "field capacity" is jargon which is fine in the paper in general, however ideally the very first paragraph should give strong accessible motivation for the paper)

P2 L18 "Other studies..." - confusing sentence

P2 L22 Are the units really kg m-2? Total water per volume suggests kg m-3. In any case, I am not sure that information on the units is really necessary here.

P2 L24 "The lack of..." This paragraph should be revised. The first sentence states a problem – though instead of "a significant issue" could you be more explicit? Following this it would help the casual reader to make it clearer that ECVs & data assimilation are potential solutions to this problem

P2 L31 "Besides, data assimilation...". 'Besides' is a strange word to use here.

P2 L22 "In particular, the assimilation of LAI..." This is a key piece of motivating research and it would help to make more of it...e.g "Previous work has studied the impact of assimilation of LAI observations and found that...".

P3 L1 "The ISBA LSM..." This paragraph describing some results specific to this model in detail is out of place in the introduction – I suggest removing and incorporating the relevant information in section 3.1.

P3 L10 "On the other hand, no more than 27%...presented significant correlations". Unnecessary elaborate use of language. A clearer way to put it would be: "On the other hand, only 27%...had significant correlations".

P3 L15 "to retrieve". Retrieve is used throughout but feels like the wrong word. "Estimate" would be more accurate

P3 L26 "IM and LT. With already a large number of acronyms in the paper, these new acronyms are unnecessary and add to confusion. As a reader I would prefer to continually read "inverse modelling" and "LDAS tuning" method, rather than the acronyms – I found it necessary to remind myself of the meaning of these terms.

P4 L9 "They highlighted that " ". Why do you quote the author talking about their results here whilst describing results yourself elsewhere? Quotations like this is highly unusual and recommend avoiding.

P4 L11 "They give the following scores..." the $R2$ values are not really informative, unless you also provide information about the spatial scale, time period (annual, monthly, daily?) that the validation was carried across. But overall I think this entire sentence

is too much information – I think it is sufficient to say that the product is well evaluated against ground observations and leave it at that. The particularly interested reader can follow the reference.

P6 equation 2. This is two equations, please split.

P6 L18 "The t superscript stands for time (t)". Adding (t) is unnecessary.

P6 L19 "The initial time (t=0) is denoted by the 0 superscript." Again, (t=0) is unnecessary.

P6 L21 "The yt term of ...". The description of these equations is slightly out of order. I would move this yt up, where you describe all the terms in the delta x equation of (2). After you have described all the terms in this equation, then add the second equation for K=..., then describe all the terms here.

P6 L22 "i.e. the model predicted value of the observation at the analysis time". I am not an expert in data assimilation, but this sounds strange. I assume you just mean "the modelled value at the analysis time". Please reread and ensure that you feel that this whole section is sufficiently precise and clear, particularly for non-experts.

P6 equation 3. h (lower case) appears to be undefined. Later on in equation 6 y(x) is used. Either a typo or missing description.

P7 L7 "The standard deviation of errors of GEOV1 is assumed to be 20% of GEOV1 LAI". Why do you make this assumption, do you have any basis? If possible, please explain your reason, or at least help the sceptical reader trust that it is reasonable.

P8 L12-14 It is not quite clear what you did here, by calculating the average B above a threshold of 90% of its maximum. Why does this limit the impact of model errors? Please explain.

P9 Section 3.5.1. This reads like bullet points, please expand to prose. P9 L7 "by minimising this cost function". This makes the optimisation sound more complicated

than it is. Preferably explain as simply as possible i.e., "the MaxAWC used in the simulation with the lowest RMSE was selected as the optimal one."

Results section generally good, though please re-read for clarity.

Discussion: the structure of the section into five questions is appealing – this approach would be improved if you start each subsection with a clear sentence which answers the question. Currently some sections start with dense recapitulation of the methods, or answers to questions different from those which are posed.

e.g. 5.1 What is the added value of the LDAS? "The LADS approach allows sequential integration of LAI observations into the model".

Instead: "The LDAS approach leads to more realistic simulations of LAI and Bag. In addition, N does not need to be determined".

Overall I would recommend editing of this section to make it more streamlined.

Section 5.6 is mislabelled (or section 5.5 is missing). Figures & tables

Suggest moving table 1 / 2 to supplementary material, or making a concise version of table 2 for the main paper and moving the rest to supplementary.

Figure 1: The caption is slightly confusing to read. For one thing, the colour of the symbols is redundant – they are uniquely determined by their shape, therefore you can precisely just use this to refer to them using just the symbol in the caption. Also, "Colour symbols show the departments presenting a significant correlation..." is confusing, when all the symbols are coloured (arguably, black is a colour). Finally, "empty blue circles" is confusing at first, since many of the circles on the plot are filled with another symbol. Suggest instead just "circles". Overall consider revising and unpacking this caption to make it clearer, and potentially revise the use of colour in the figure. Potentially the figure could be reproduced using just a single colour for all symbols without any loss of precision.

Figure 3: figures too small, would be better if they were placed in a 2x2 panel plot and resized.

Figures 4,5,8,9,10,12 could each be placed on a single row with two figures, rather than a single column. Would help fit nicer on a page. Some could also be combined (e.g. 8 & 9, or 11 & 12).

―――――――――――――――――――

---

## Referee Comment (RC2) · Anonymous Referee #2 · 24 May 2017

This study uses satellite-derived low resolution Leaf Area Index (LAI) to estimate Soil Maximum Available Water Content (MaxAWC) with the overarching goal of improving representation of drought. The optimal value of MaxAWC is estimated by using two different methods (i) a simple inverse modelling technique and (ii) a Land Data Assimilation System (LDAS). LDAS results in better and more realistic estimates of MaxAWC than the simpler inverse modeling technique. The study fits very well within the scope of the journal HESS. It is technically sound and well structured. I do have a few comments though which need to be addressed before I can recommend publication.

Major comment:

(1) As of now the authors validate the drought representation of the model by comparing the annual maximum above-ground biomass (Bag) and straw cereal grain yield (GY) values only. In my opinion for better drought representation, it is also important to see how the selection of MaxAWC influences drought representation in terms of water balance (ET, Runoff, Soil Moisture). This would also provide an independent criterion for model evaluation for drought representation. The authors may want to use observations such as streamflow, satellite based SM or ET for the evaluation purposes.

(2) The introduction section needs to be improved by ensuring a better connection between the focus of a paragraph with the one following it. For example, as of now the paragraph two (starting on line 5 page 2) seems out of place. The paragraphs before and after it discuss the influence of MaxAWC and this one discusses the influence of climate variability. Likewise, the discussion of data assimilation starting on line 30 page 2, also seems to be out of place.

Minor comments: (1) Line 23 (page 1): Not just due to climate change, but in the context of natural climate variability too. (2) Line 2 (page 2): Almost all regions are affected by drought, it's just some are more sensitive/vulnerable to drought risks exposure than the others. (3) Page 2, Line 5: "Assigning agricultural. . .. . ." rephrase this sentence for better clarity, please. (4) Page 2 Line 8: "Li et al. (2010) showed. . .." Please provide an estimate of the scales here. (5) Page 2 Line 12: Please change this sentence to: "Soil characteristic influence the vegetation response to. . ...". (6) Page 2 line 12: Please change "In the model benchmarking study of Eitzinger et al. (2004)," to "In a model benchmarking study, Etizinger et al., (2004) . . .." (7) Page 2, Line 14: Please change "differing" to "that differ". (8) Page 2, Line 17: Please change "taking into account soil type" to "taking into account of soil type". (9) Page 8, Line 2, "Of" is missing in "relevance the". (10) Page 8, Line 11: Please change "consists in" to "consists of". (11) Caption of Figure 4: "Dark" should be "black".

---

## Author Comment (AC1) · 19 Jun 2017

RESPONSE TO REVIEWER #1

The authors thank anonymous reviewer 1 for his/her review of the manuscript and for the fruitful comments.

1.1 [The work is technically sound, scientifically interesting and worthy of publication. However I do suggest some revisions to the text for clarity and readability and beyond these specific revisions recommend further proof reading by the authors, a native English speaker and/or the journals editorial team. Particular attention should be paid

to clarity in the introduction as improvements here would encourage more readers to engage with the paper.]

Response 1.1:

Yes, we will re-read the whole paper. Should a revised version of this paper be accepted in HESS, a copy editing work will be performed.

1.2 [P1 L8 "this parameter is usually unavailable" - slightly awkward, perhaps "this parameter is uncertain".]

Response 1.2:

Agreed.

1.3 [P1 L23 "supervision" - not sure what is meant by this.]

Response 1.3:

Yes. "There is a need for better supervision of the impacts of droughts" was replaced by "There is a need to monitor the impacts of droughts".

1.4 [P1 L29 "This quantity..." - this sentence is confusing and could be improved; please bear in mind any non-expert readers (e.g. "field capacity" is jargon which is fine in the paper in general, however ideally the very first paragraph should give strong accessible motivation for the paper]

Response 1.4:

Yes. "at field capacity" was deleted.

1.5 [P2 L18 "Other studies..." - confusing sentence.]

Response 1.5:

Yes. This sentence was reworded as: "Tanaka et al. (2004), Portoghese et al. (2008), and Piedallu et al. (2011), have highlighted the important role of the soil characteristics

(soil texture, rooting depth) on MaxAWC. Soylu et al. (2011) and Wang et al. (2012) illustrated the major impact of MaxAWC on evapotranspiration."

1.6 [P2 L22 Are the units really kg m-2? Total water per volume suggests kg m-3. In any case, I am not sure that information on the units is really necessary here..]

Response 1.6:

Yes. The sentence was reworded as: "While soil properties such as soil texture determine the soil water holding capacity (in kg m-3), information on rooting depth is needed to determine MaxAWC (in kg m-2)."

1.7 [P2 L24 "The lack of..." This paragraph should be revised. The first sentence states a problem – though instead of "a significant issue" could you be more explicit? Following this it would help the casual reader to make it clearer that ECVs & data assimilation are potential solutions to this problem.]

Response 1.7:

Yes. The sentence was reworded as: "The lack of in situ observations of MaxAWC to calibrate and assess LSMs impacts the ability of LSMs to represent drought effects on plants. Using satellite observations and data assimilation techniques could be a solution to this problem".

1.8 [P2 L31 "Besides, data assimilation...". 'Besides' is a strange word to use here..]

Response 1.8:

Yes. "Besides" was deleted.

1.9 [P2 L22 "In particular, the assimilation of LAI..." This is a key piece of motivating research and it would help to make more of it...e.g "Previous work has studied the impact of assimilation of LAI observations and found that...".]

Response 1.9:

Yes. The sentence was reworded as: "Previous work has studied the impact of assimilation of LAI observations and found that it can significantly improve the representation of vegetation growth (e.g. Albergel et al., 2010 ; Barbu et al., 2011, 2014)."

1.10 [P3 L1 "The ISBA LSM..." This paragraph describing some results specific to this model in detail is out of place in the introduction – I suggest removing and incorporating the relevant information in section 3.1..]

Response 1.10:

We think that this paragraph is needed in the Introduction to present the rationale for the present study.

1.11 [P3 L10 "On the other hand, no more than 27%...presented significant correlations". Unnecessary elaborate use of language. A clearer way to put it would be: "On the other hand, only 27%...had significant correlations"..]

Response 1.11:

Yes. The sentence was reworded as: "On the other hand, only 27% of the 45 straw cereals départements (i.e. only 12 départements) had significant correlations".

1.12 [P3 L15 "to retrieve". Retrieve is used throughout but feels like the wrong word. "Estimate" would be more accurate]

Response 1.12:

Yes. Throughout the text, "retrievals" was replaced by "estimates", and "to retrieve" was replaced by "to estimate".

1.13 [P3 L26 "IM and LT. With already a large number of acronyms in the paper, these new acronyms are unnecessary and add to confusion. As a reader I would prefer to continually read "inverse modelling" and "LDAS tuning" method, rather than the acronyms – I found it necessary to remind myself of the meaning of these terms.]

Response 1.13:

Agreed.

1.14 [P4 L9 "They highlighted that". Why do you quote the author talking about their results here whilst describing results yourself elsewhere? Quotations like this is highly unusual and recommend avoiding.]

Response 1.14:

This paragraph was moved to the Supplement.

1.15 [P4 L11 "They give the following scores..." the R2 values are not really informative, unless you also provide information about the spatial scale, time period (annual, monthly, daily?) that the validation was carried across. But overall I think this entire sentence is too much information – I think it is sufficient to say that the product is well evaluated against ground observations and leave it at that. The particularly interested reader can follow the reference.]

Response 1.15:

Yes. This paragraph was moved to the Supplement and replaced by: "The product is well evaluated against ground observations (see the Supplement)".

1.16 [P6 equation 2. This is two equations, please split.]

Response 1.16:

Agreed.

1.17 [P6 L18 "The t superscript stands for time (t)". Adding (t) is unnecessary]

Response 1.17:

Agreed.

1.18 [P6 L19 "The initial time (t=0) is denoted by the 0 superscript." Again, (t=0) is

unnecessary.]

Response 1.18:

Agreed.

1.19 [P6 L21 "The yt term of ...". The description of these equations is slightly out of order. I would move this yt up, where you describe all the terms in the delta x equation of (2). After you have described all the terms in this equation, then add the second equation for K=..., then describe all the terms here.]

Response 1.19:

Agreed.

1.20 [P6 L22 "i.e. the model predicted value of the observation at the analysis time". I am not an expert in data assimilation, but this sounds strange. I assume you just mean "the modelled value at the analysis time". Please reread and ensure that you feel that this whole section is sufficiently precise and clear, particularly for non-experts.]

Response 1.20:

Agreed.

1.21 [P6 equation 3. h (lower case) appears to be undefined. Later on in equation 6 y(x) is used. Either a typo or missing description.]

Response 1.21:

Yes. "h" is now defined as the observation operator.

1.22 [P7 L7 "The standard deviation of errors of GEOV1 is assumed to be 20% of GEOV1 LAI". Why do you make this assumption, do you have any basis? If possible, please explain your reason, or at least help the sceptical reader trust that it is reasonable.]

Response 1.22:

Yes. This sentence was reworded as: "This assumption is based on option 3 presented in Barbu et al. (2011). They showed that this option gives the best simulated LAI over an instrumented grassland site in southwestern France".

1.23 [P8 L12-14 It is not quite clear what you did here, by calculating the average B above a threshold of 90% of its maximum. Why does this limit the impact of model errors? Please explain.]

Response 1.23:

Yes. The following sentences were added in Sect. 3.4: "In drought conditions, modelled Bag can rise to a maximum value and then drop rapidly. Therefore the peak Bag can be dependent on modelling uncertainties and on uncertainties in the atmospheric forcing".

1.24 [P9 Section 3.5.1. This reads like bullet points, please expand to prose. P9 L7 "by minimising this cost function". This makes the optimisation sound more complicated than it is. Preferably explain as simply as possible i.e., "the MaxAWC used in the simulation with the lowest RMSE was selected as the optimal one."]

Response 1.24:

Agreed.

1.25 [Results section generally good, though please re-read for clarity. Discussion: the structure of the section into five questions is appealing – this approach would be improved if you start each subsection with a clear sentence which answers the question. Currently some sections start with dense recapitulation of the methods, or answers to questions different from those which are posed. e.g. 5.1 What is the added value of the LDAS? "The LDAS approach allows sequential integration of LAI observations into the model". Instead: "The LDAS approach leads to more realistic simulations of LAI and Bag. In addition, N does not need to be determined". Overall I would recommend editing of this section to make it more streamlined.]

Response 1.25:

Yes. We reorganised the Discussion sub-sections accordingly.

1.26 [Section 5.6 is mislabelled (or section 5.5 is missing).]

Response 1.26:

Yes. This correction was made.

1.27 [Suggest moving table 1 / 2 to supplementary material, or making a concise version of table 2 for the main paper and moving the rest to supplementary.]

Response 1.27:

Agreed.

1.28 [Figure 1: The caption is slightly confusing to read. For one thing, the colour of the symbols is redundant – they are uniquely determined by their shape, therefore you can precisely just use this to refer to them using just the symbol in the caption. Also, "Colour symbols show the departments presenting a significant correlation..." is confusing, when all the symbols are coloured (arguably, black is a colour). Finally, "empty blue circles" is confusing at first, since many of the circles on the plot are filled with another symbol. Suggest instead just "circles". Overall consider revising and unpacking this caption to make it clearer, and potentially revise the use of colour in the figure. Potentially the figure could be reproduced using just a single colour for all symbols without any loss of precision.]

Response 1.28:

Yes. "yellow down triangle" was replaced by "green down triangle". This improved the readability of Fig. 1.

1.29 [Figure 3: figures too small, would be better if they were placed in a 2x2 panel plot and resized. Figures 4,5,8,9,10,12 could each be placed on a single row with two figures, rather than a single column. Would help fit nicer on a page. Some could also be combined (e.g. 8 & 9, or 11 & 12).]

[Figure]

Response 1.29:

We prefer leaving the Figure layout as is. We think it will facilitate the inclusion of the Figures in the two-column format of HESS.

---

## Author Comment (AC2) · 19 Jun 2017

RESPONSE TO REVIEWER #2

The authors thank anonymous reviewer 2 for his/her review of the manuscript and for the fruitful comments.

2.1 [As of now the authors validate the drought representation of the model by comparing the annual maximum above-ground biomass (Bag) and straw cereal grain yield (GY) values only. In my opinion for better drought representation, it is also important to see how the selection of MaxAWC influences drought representation in terms of

water balance (ET, Runoff, Soil Moisture). This would also provide an independent criterion for model evaluation for drought representation. The authors may want to use observations such as streamflow, satellite based SM or ET for the evaluation purposes.]

Response 2.1:

Using independent satellite-derived products for validation is a very good idea but some limitations have to be considered. We made an attempt to use the GLEAM evapotranspiration product (Miralles et al., 2011) but very poor correlations were obtained for most départements (median R2 values less than 0.06). Using streamflow observations would require the coupling with an hydrological model. This is out of the scope of this study. On the other hand, good correlations were found for the Gross Primary Production (GPP) FLUXNET-MTE product described in Jung et al. (2009). With respect to basic ISBA simulations, GPP RMSE is nearly systematically improved by the original LDAS simulations, and LDAS tuning drastically reduces the largest RMSE values. A new Figure presenting these results will be added.

References:

Jung, M., Reichstein, M., and Bondeau, A.: Towards global empirical upscaling of FLUXNET eddy covariance observations: validation of a model tree ensemble approach using a biosphere model, Biogeosciences, 6, 2001–2013, doi:10.5194/bg-6-2001-2009, 2009.

Miralles, D. G., Holmes, T. R. H., De Jeu, R. A. M., Gash, J. H., Meesters, A. G. C. A., and Dolman, A. J.: Global land-surface evaporation estimated from satellite-based observations, Hydrology and Earth System Sciences, 15, 453–469, doi:10.5194/hess-15-453-2011, 2011.

2.2 [The introduction section needs to be improved by ensuring a better connection between the focus of a paragraph with the one following it. For example, as of now the paragraph two (starting on line 5 page 2) seems out of place. The paragraphs before

and after it discuss the influence of MaxAWC and this one discusses the influence of climate variability. Likewise, the discussion of data assimilation starting on line 30 page 2, also seems to be out of place.]

Response 2.2:

Yes. Paragraph 2 was moved before the first paragraph. Data assimilation is now introduced before as: "Using satellite observations and data assimilation techniques could be a solution to this problem."

2.3 [(1) Line 23 (page 1): Not just due to climate change, but in the context of natural climate variability too.]

Response 2.3:

Agreed.

2.4 [(2) Line 2 (page 2): Almost all regions are affected by drought, it's just some are more sensitive/vulnerable to drought risks exposure than the others.]

Response 2.4:

Yes. "In regions affected by drought" was replaced by "In regions vulnerable to drought risk exposure,"

2.5 [(3) Page 2, Line 5: "Assigning agricultural..." rephrase this sentence for better clarity, please.]

Response 2.5:

Yes. "Assigning" was replaced by "Comparing".

2.6 [(4) Page 2 Line 8: "Li et al. (2010) showed: : :." Please provide an estimate of the scales here]

Response 2.6:

Yes. This sentence was changed to: "Li et al. (2010) showed that air temperature tends to influence mean crop yields at small scales (400 to 600 km) whereas rainfall drives crop yields at larger scales (50 to 300 km)".

2.7 [(5) Page 2 Line 12: Please change this sentence to: "Soil characteristic influence the vegetation response to...".]

Response 2.7:

Agreed.

2.8 [(6) Page 2 line 12: Please change "In the model benchmarking study of Eitzinger et al. (2004)," to "In a model benchmarking study, Etizinger et al., (2004) ..."]

Response 2.8:

Agreed.

2.9 [(7) Page 2, Line 14: Please change "differing" to "that differ".]

Response 2.9:

Agreed.

2.10 [(8) Page 2, Line 17: Please change "taking into account soil type" to "taking into account of soil type".]

Response 2.10:

Agreed.

2.11 [(9) Page 8, Line 2, "Of" is missing in "relevance the".]

Response 2.11:

Yes. this was corrected.

2.12 [(10) Page 8, Line 11: Please change "consists in" to "consists of".]

Response 2.12:

Agreed.

2.13 [(11) Caption of Figure 4: "Dark" should be "black".]

Response 2.13:

Agreed.